# Foxtrot migration and dynamic over-wintering range of an Arctic raptor

Ivan Pokrovsky[1]*, Teja Curk[2], Andreas Dietz[3], Ivan Fufachev[4], Olga Kulikova[5], Sebastian Rößler[3], Martin Wikelski[1,6]

[1]Department of Migration, Max Planck Institute of Animal Behavior, Radolfzell, Germany; [2]Leibniz Institute for Zoo and Wildlife Research, Berlin, Germany; [3]Deutsches Zentrum für Luft- und Raumfahrt e.V. (DLR), Wessling, Germany; [4]Institute of Plant and Animal Ecology, Yekaterinburg, Russian Federation; [5]Institute of the Biological Problems of the North, Magadan, Russian Federation; [6]Centre for the Advanced Study of Collective Behaviour, University of Konstanz, Konstanz, Germany

## eLife assessment

This **fundamental** work describes an understudied bird migration pattern using data from an Arctic raptor. With an extensive dataset and comprehensive analyses, the observed pattern is **convincing**. This study will be of interest to researchers exploring the ecological drivers of bird migration.

*For correspondence:
ipokrovsky@ab.mpg.de

Competing interest: The authors declare that no competing interests exist.

**Abstract** Advances in tracking technologies have revealed the diverse migration patterns of birds, which are critical for range mapping and population estimation. Population trends are usually estimated in breeding ranges where birds remain stationary, but for species that breed in remote areas like the Arctic, these trends are often assessed in over-wintering ranges. Assessing population trends during the wintering season is challenging due to the extensive movements of birds in these ranges, which requires a deep understanding of the movement dynamics. However, these movements remain understudied, particularly in the mid-latitudes, where many Arctic breeders overwinter, increasing uncertainty in their ranges and numbers. Here, we show that the Arctic breeding raptor Rough-legged buzzard, which overwinters in the mid-latitudes, has a specific wintering strategy. After migrating ca. 1500 km from the Arctic to mid-latitudes, the birds continue to move throughout the entire over-wintering period, traveling another 1000 km southwest and then back northeast as the snowline advances. This continuous movement makes their wintering range dynamic throughout the season. In essence, this movement represents an extension of the quick migration process, albeit at a slower pace, and we have termed this migration pattern 'foxtrot migration', drawing an analogy to the alternating fast and slow movements of the foxtrot dance. These results highlight the potential errors in range mapping from single mid-winter surveys and emphasize the importance of this migration pattern in assessing the conservation status of bird species. Understanding this migration pattern could help to correctly estimate bird populations in over-wintering ranges, which is especially important for species that nest in hard-to-reach regions such as the Arctic.

## Introduction

In recent years, advances in tracking technology have greatly improved our understanding of bird migration (*Wikelski et al., 2007*; *Kays et al., 2015*; *Jetz et al., 2022*; *Kays and Wikelski, 2023*). By tracking tagged birds, we can study various migratory behaviors, such as their propensity to migrate, connectivity and philopatry, length of migration routes, variations in flight speed, and movements within a single season (*Alerstam et al., 2003*; *Newton, 2008*; *Alerstam, 2011*; *Berthold et al., 2013*;

*Trierweiler et al., 2013*; *Schlaich et al., 2023*). Different combinations of these behaviors contribute to various bird migration patterns (*Chapman et al., 2014*; *Alerstam and Bäckman, 2018*; *Lislevand et al., 2020*). Understanding the migratory patterns of birds is critical to accurately map their ranges and estimate their population trends. Typically, population trends are assessed in breeding areas, where birds are linked to their nesting sites, allowing reliable estimates of breeding numbers (*Reif, 2013*). However, access to some breeding areas, such as the Arctic, presents logistical and political challenges (*Gallo-Cajiao et al., 2023*; *Koivurova and Shibata, 2023*). As a result, population trends for these species are often estimated in over-wintering ranges where birds are not tied to specific locations (*Robinson et al., 2005*; *Jia et al., 2016*). This challenges accurately censusing birds and delineating their ranges, especially when birds move extensively in over-wintering ranges during the season, and censuses are conducted only once per over-wintering season. Therefore, understanding bird movements in over-wintering ranges is essential for understanding the ecology of birds in these regions and has practical implications for accurately assessing their ranges and trends in their populations.

Many bird species experience significant seasonal changes in food availability within their over-wintering range due to environmental conditions. This often triggers movements across these areas. For Palearctic migrants wintering in Africa, these movements have been studied extensively (*Moreau, 1972*; *Trierweiler et al., 2013*; *Thorup et al., 2017*; *Schlaich et al., 2023*). However, the over-wintering movements of mid-latitude birds remain poorly understood. At the same time, many Arctic birds overwinter in the mid-latitudes, and understanding their movement patterns could be crucial for interpreting their population trends. In the mid-latitudes, food availability is strongly influenced by environmental factors such as snow cover dynamics. The progressive movement of the snow cover line from north (-east) to south (-west) and back again between October and May in many mid-latitude regions, particularly those exposed to northwesterly wind systems, significantly affects food availability for numerous bird species (*Sonerud, 1986*; *Vansteelant et al., 2011*). We propose that species dependent on this environmental factor gradually move away from snow-covered areas during winter and then gradually move back in the opposite direction, resulting in a directed and continuous displacement of their over-wintering range.

The Rough-legged buzzard (*Buteo lagopus*) is an Arctic breeding and mid-latitude wintering raptor (*Ferguson-Lees and Christie, 2001*; *Bechard and Swem, 2002*). Rough-legged buzzards feed mainly on small rodents in the Arctic and its wintering grounds (*Tast et al., 2010*; *Pokrovsky et al., 2014*). They prefer open areas for hunting, and trees and tall bushes or uplands for resting and nesting. In the Arctic, where they nest, these habitats are in the southern and typical tundra zone (*Walker et al., 2005*). In the mid-latitudes, where they overwinter, these habitats are in the zone of fields with scattered patches of forests (wooded fields). The taiga zone, located between the tundra and wooded fields zones, offers limited open space, and is generally unsuitable for the species, although they may nest in the northern taiga near the tundra zone (*Sundell et al., 2004*). Thus, Rough-legged buzzards migrate from the Arctic through the taiga zone to the wooded fields zone for wintering. However, environmental conditions in the mid-latitudes vary, potentially influencing their migratory behavior.

The spatio-temporal latitudinal and longitudinal gradients in environmental conditions impact bird populations (*Kiat et al., 2021*). Deep snow covers the northeastern regions of suitable wintering habitats (wooded fields) during mid-winter, significantly hindering the Rough-legged buzzards' ability to hunt small rodents (*Sonerud, 1986*; *Vansteelant et al., 2011*; *Terraube et al., 2015*; *Curk et al., 2020*). Simultaneously, many Arctic migrants prefer winter as close to the Arctic as possible to shorten their return to breeding grounds and extend the breeding season (*Berthold et al., 2013*). Therefore, we can anticipate three potential non-exclusive migration strategies for Rough-legged buzzards: (1) birds may migrate through the taiga and northeastern wooded fields zones to the southwestern areas with less snow cover during winter, (2) birds may stop in the wooded fields zone after crossing the taiga and move southwest in mid-winter when snow cover makes hunting rodents impossible, eventually migrating back to the breeding grounds from southwestern regions, (3) alternatively, similar to the second strategy, birds may move southwest but return to the northeastern wooded fields zone after mid-winter to be closer to the breeding grounds, and migrating to the Arctic from there.

In this study, we tracked GPS-tagged Rough-legged buzzards for 10 years and analyzed their movements in the over-wintering mid-latitude range in relation to snow cover. For this study, we made the following predictions. (1) Rough-legged buzzards could exhibit a directional and seasonal movement

pattern during the over-wintering period, moving from the northeast to the southwest and back again. This would result in a dynamic over-wintering range that would continue to move geographically throughout the season. (2) The over-wintering movements would differ from the fall and spring migrations in duration, extent, speed, and direction but would continue throughout the season. Thus, over the entire annual cycle, Rough-legged buzzards would be characterized by a migratory pattern consisting of an alternation of quick (fall and spring migrations) and slow (over-wintering movements) phases. (3) Over-wintering movements would occur in suitable open habitats, while fall and spring migrations would occur in unsuitable forested areas. Thus, vegetation land cover will determine the extent of quick and slow migration phases. (4) During over-wintering movements, Rough-legged

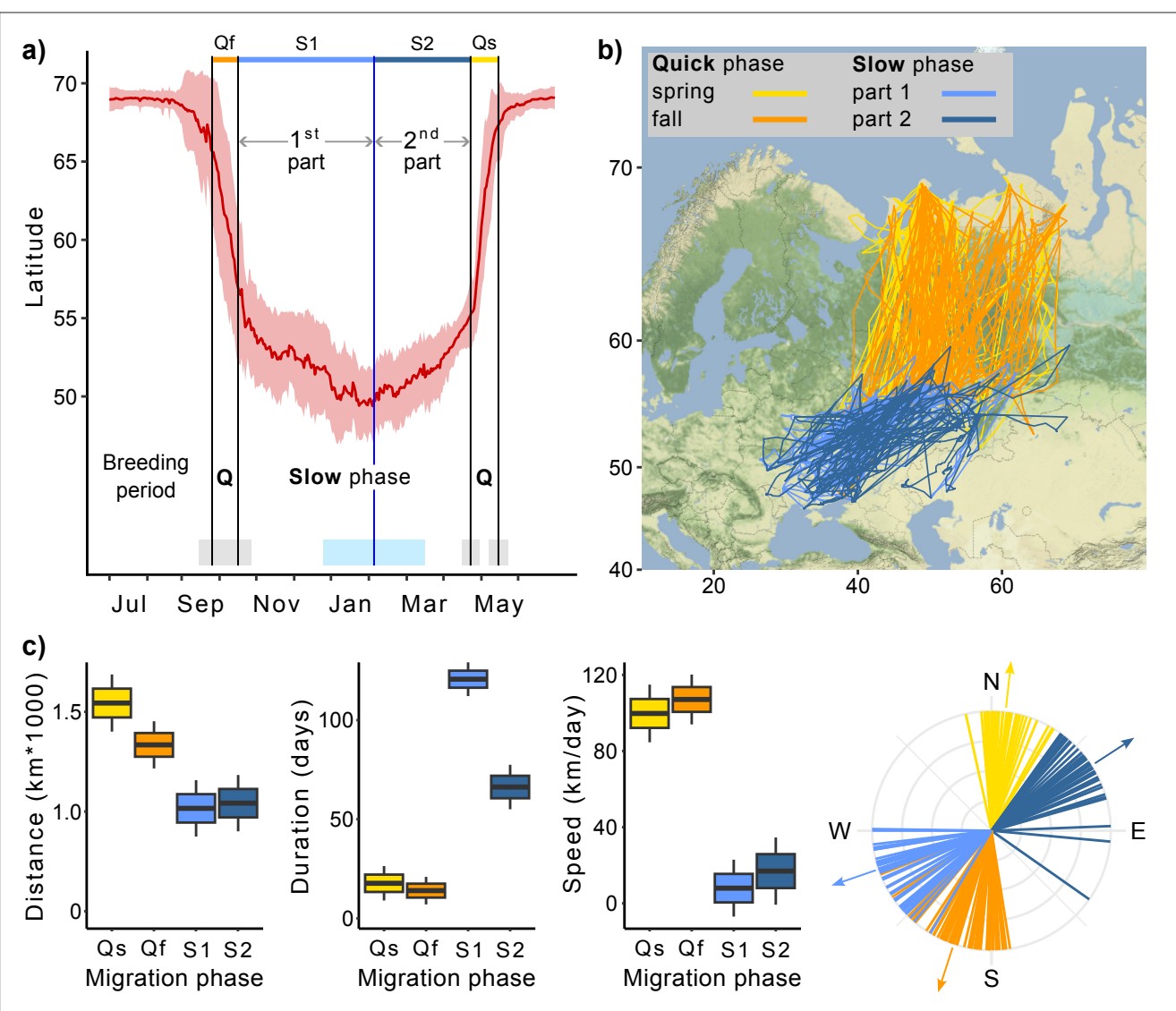

**Figure 1.** Migration of Rough-legged buzzards. Q, quick phase; Qf, quick fall phase (orange); Qs, quick spring phase (yellow); S1, slow phase, first part (light blue); S2, slow phase, second part (dark blue). (**a**) Change in the latitude of 43 Rough-legged buzzards during the year, red line indicates mean latitude of all birds, black vertical lines indicate mean dates of start and end of the migration phases, blue vertical line indicates mean date of the minimum latitude. Gray, sky blue, and piggy pink shaded areas indicate standard deviation of the means. (**b**) Migration map. (**c**) Difference in the migration parameters between the migration phases. Lines on the direction plot (down, right) represent the mean value for each bird; arrows represent the mean direction for each phase. Boxes on the boxplots show the interquartile range, the whiskers are maximum and minimum values.

The online version of this article includes the following figure supplement(s) for figure 1:

**Figure supplement 1.** The distribution of latitudes of Rough-legged buzzards in different months.

**Table 1.** Parameters of the Rough-legged buzzards' migration (mean ± sd).

| Phase of migration | Sub-phase of migration | Distance (km) | Duration (days) | Speed (km/day) | Direction (°) |
|---|---|---|---|---|---|
| | | 1415 ± 50 | 15 ± 3 | 104 ± 6 | |
| | Spring | 1544 ± 72 | 18 ± 4 | 100 ± 8 | 7 ± 2 |
| Quick | Fall | 1334 ± 59 | 14 ± 4 | 107 ± 7 | 198 ± 3 |
| | | 1026 ± 55 | 100 ± 4 | 12 ± 7 | |
| | First phase | 1016 ± 71 | 121 ± 4 | 8 ± 7 | 251 ± 3 |
| Slow | Second phase | 1042 ± 71 | 66 ± 6 | 17 ± 9 | 57 ± 2 |

buzzards will experience less snow cover than if they had stayed where they arrived at the end of the fall migration. Thus, snow cover dynamics will determine the dynamics of over-wintering movements.

## Results

### Migration of Rough-legged buzzards: Always on the move

Except during the breeding season, Rough-legged buzzard migration continues throughout the year, even after the birds' arrival at their traditionally recognized 'wintering grounds' (*Figure 1a and b*). Rough-legged buzzards started their fall migration on 28 September (hereafter mean ± sd for the day of the year: 271 ± 11, n = 31) and ended on 12 October (285 ± 11, n = 33). The mean latitude/longitude where the birds ended their fall migration was 55.57 ± 1.92°/49.35 ± 5.63° (*Figure 1a*). During the winter, birds continued to migrate at a slower pace down to 49.53 ± 2.01° latitude (on 5 February, 36 ± 40, n = 23) and 34.29 ± 5.11° longitude (on 24 January, 24 ± 47, n = 23). Afterward, during the second part of the winter, birds returned to 55.52 ± 2.63° latitude and 49.79 ± 8.24° longitude to start the spring migration (*Figure 1a*). Rough-legged buzzards started their spring migration to the Arctic on 27 April (117 ± 7, n = 27) and arrived at the breeding grounds on 15 May (135 ± 8, n = 18).

In the following, we will refer to the spring and fall migrations as the quick phase, the over-wintering movement to the lowest point of latitude as the first part of the slow phase, and the movement from the lowest point of latitude to the starting point of the spring migration as the second part of the slow phase (*Figure 1a*). For both quick and slow phases of the migration, linear mixed-effects models with the season as a fixed factor received higher support from the likelihood ratio test (p<0.001, *Supplementary file 1a–c*).

### Quick-slow phase features comparison

During the quick phase, individual birds flew greater distances in a shorter time, that is, at a faster rate, than during the slow phase. After arriving at what is traditionally known as the wintering grounds, the direction of migration changed, so the direction of quick and slow phases also differed (*Table 1*, *Figure 1c*). The quick phase (one part) was 1415 ± 50 km long (hereafter mean ± sd), whereas the slow phase (one part) was 1026 ± 55 km, that is, 389 ± 60 km shorter (p<0.001, *Supplementary file 1d*, *Figure 1c*). During the quick phase, birds flew for 15 ± 3 days, and one part of the slow phase lasted 100 ± 4 days, that is, 85 ± 5 days longer (p<0.001, *Supplementary file 1e*, *Figure 1c*). At the same time, the second part of the slow phase was 54 ± 7 days shorter than the first (p<0.001, *Supplementary file 1e*, *Figure 1c*). The migration speed was 104 ± 6 km/day during the quick phase and 12 ± 7 km/day during the slow phase, that is, about eight times higher (p<0.001, *Supplementary file 1f*, *Figure 1c*). During the fall migration, birds moved in the SSW direction (198 ± 3°), then turned 50 ± 3° (p<0.001, *Supplementary file 1g*, *Figure 1c*) to the west and started their first slow phase until midwinter. After that, they turned back to the NEE direction (57 ± 2°) and performed their second slow phase for several months until they turned 54 ± 3° (p<0.001, *Supplementary file 1g*, *Figure 1c*) to the north and started their spring migration. We found no significant difference between the migration distances of males and females (*Supplementary file 1h*).

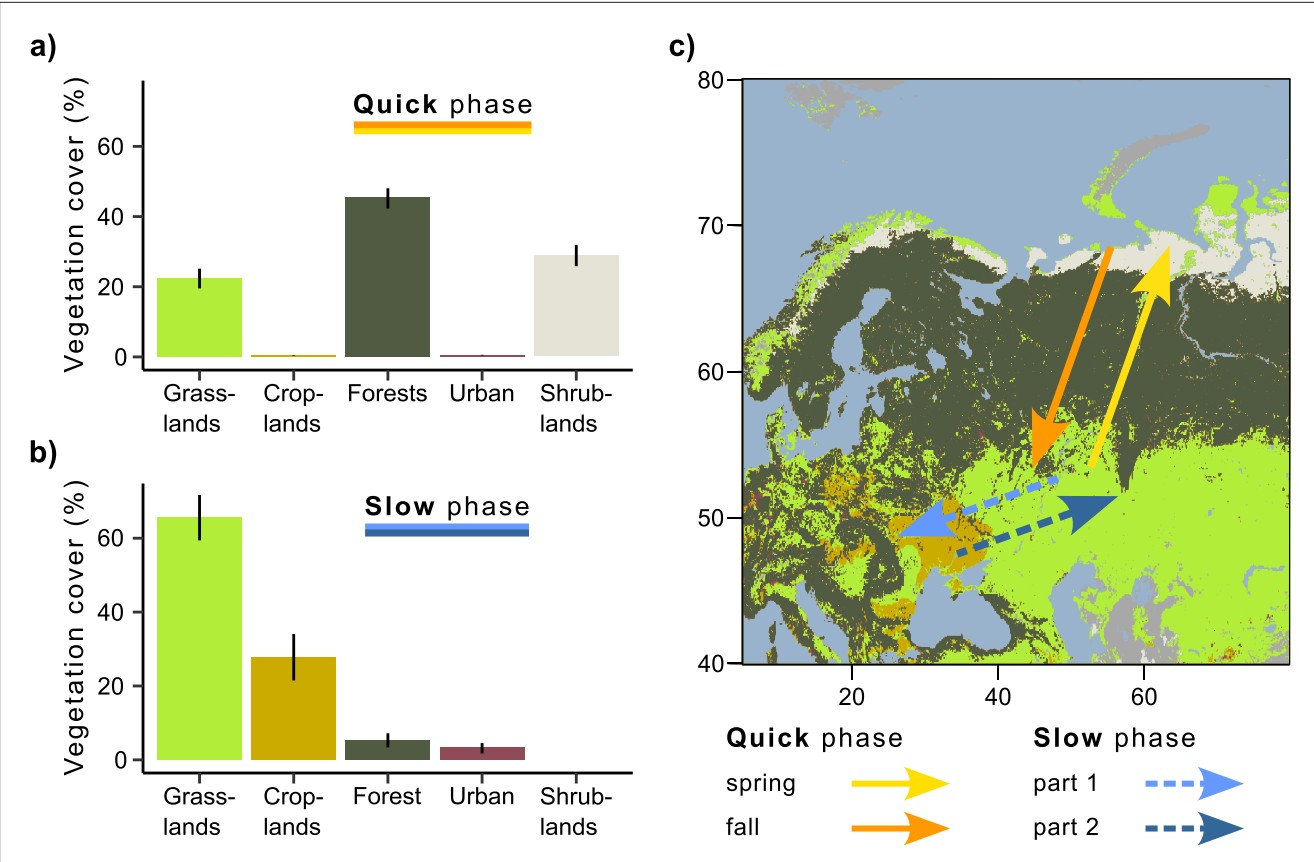

**Figure 2.** Vegetation land cover during quick and slow phases of the migration. (**a**) Quick phase (spring and fall periods together). (**b**) Slow phase (first and second parts together). On both (**a**) and (**b**), the bars show the percentage of all mean daily positions annotated with the given vegetation type ± sd. (**c**) Migration map.

## Vegetation land cover during migration

During quick phases, Rough-legged buzzards crossed the forest zone, while during the slow phase, they migrated within the grassland and cropland zones (*Figure 2*). Rough-legged buzzards migrated fast across the tundra zone on the north in the Arctic and then through the taiga zone. Therefore, during the quick phase, the three most common vegetation land cover types were forest (44.5 ± 2.9 %, hereafter, percentage of all mean daily positions annotated with the given vegetation type ± sd), shrublands (29.9 ± 3 %), and grasslands (24.7 ± 2.8 %, *Figure 2a*). During the slow phase, the three most common vegetation land cover types were grasslands (65.1 ± 6.2 %), croplands (26.9 ± 6.3 %), and forests (4.9 ± 1.4 %, *Figure 2b*). According to the linear mixed-effects models, the percentage of all vegetation land cover types differed between the slow and quick phases (p<0.001, *Supplementary file 1i*), except for the urban lands. Urban lands were more common during the slow than quick phase (*Figure 2*). However, this type has been annotated for too few birds to make an adequate comparison.

## Dynamic winter range

During the slow phase of the migration, Rough-legged buzzards experienced snow cover ranging from 4.8 ± 1.0% in October (hereafter mean ± sd) to 85.2 ± 4.6% in February (*Figure 3*). If birds spent the winter in the place where they arrived after the fall migration, they would experience snow cover conditions ranging from 4.6 ± 0.6% in October to 99.5 ± 0.1% in February (*Figure 3b*, green line). If birds fly directly to the southwest and stay there for the whole winter, they would experience snow cover conditions ranging from 1.4 ± 0.2% in October to 81.1 ± 5.0% in January (*Figure 3b*, red line). Thus, if birds fly immediately to the southwest and stay there until the end of the winter, they will find conditions with less snow cover in spring (p<0.001, *Supplementary file 1j*). And if birds stay where they ended the fall migration, they will find themselves in situations with more snow cover (p<0.001,

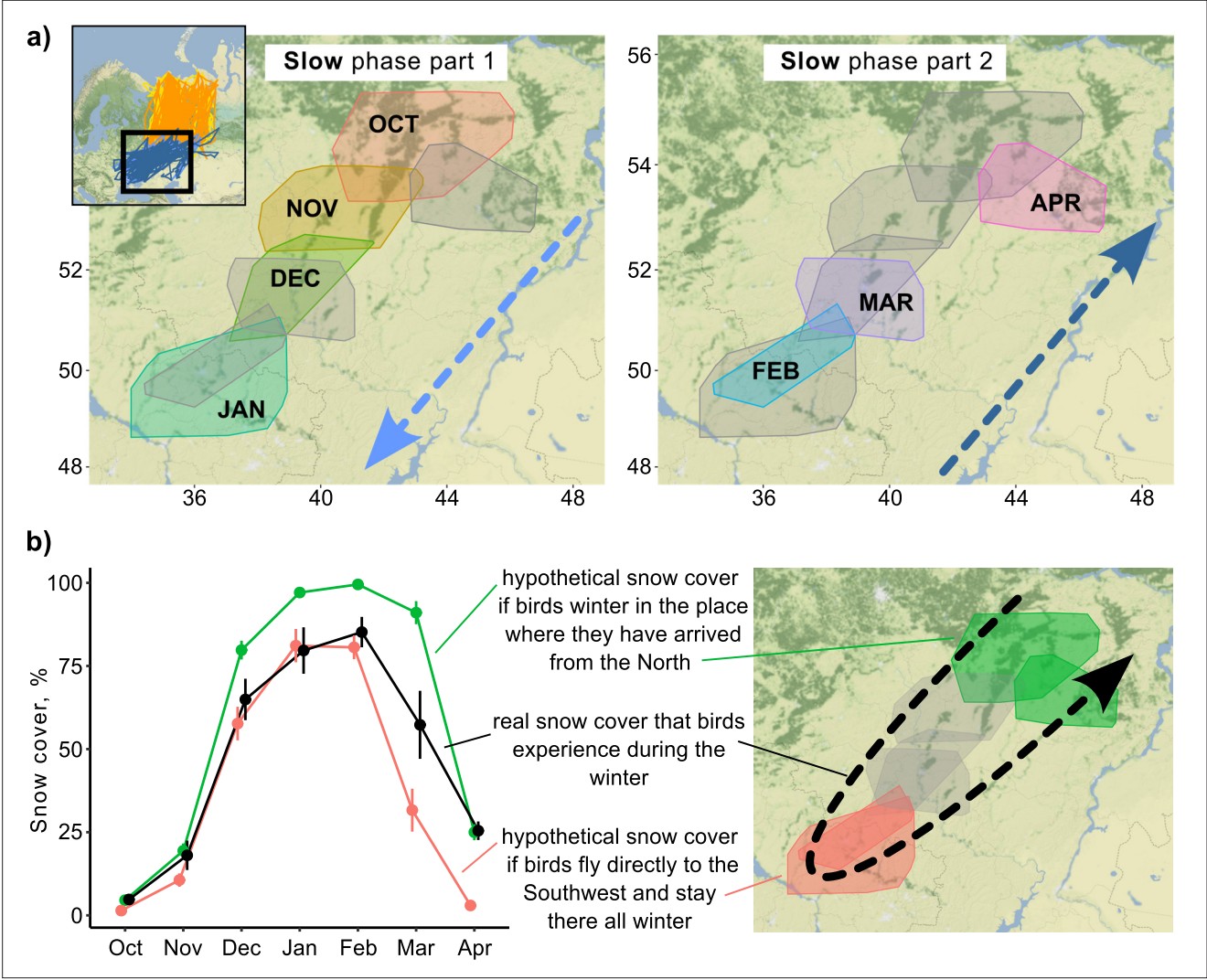

**Figure 3.** Snow cover conditions during the slow phase of the migration. (**a**) 95% minimum convex polygons (MCPs) of Rough-legged buzzards during winter. Arrows indicate the direction of the movement across months. OCT, October; NOV, November; DEC, December; JAN, January; FEB, February; MAR, March; APR, April. (**b**) Snow cover conditions for the real situation (black) and two hypothetical situations – if birds spend the winter in the place where they arrived after the fall migration (green) and if birds fly directly to the southwest and stay there all winter (red). Dots represent mean values, error lines indicate standard deviations.

The online version of this article includes the following figure supplement(s) for figure 3:

**Figure supplement 1.** The distance between two consecutive monthly minimum convex polygons (MCPs) during the over-wintering period was not influenced by (**a**) snow cover extent on the first MCP (p=0.45) or (**b**) the difference in snow cover between two consecutive monthly MCPs (p=0.36).

*Supplementary file 1j*). In the latter case, the difference between real and hypothetical situations is not as pronounced (85.2% vs. 99.5%), but it means that snow cover will be close to 100% for several months in this hypothetical situation (*Figure 3b*). The distance between two consecutive monthly minimum convex polygons (MCPs) during the over-wintering period did not depend on the snow cover extent (p=0.45) or the difference in snow cover (p=0.36, *Figure 3—figure supplement 1*).

## Materials and methods
### Dataset
For this study, we tracked 43 adult Rough-legged buzzards (35 females and 8 males) with the solar GPS-GSM loggers (E-obs GmbH and UKn – University of Konstanz). Here, 28 birds were fitted with E-obs loggers, 13 birds were fitted with UKn loggers, and 2 birds were initially fitted with UKn loggers

that were later replaced with E-obs loggers. E-obs loggers weighed 45 g, or 3.6% of the bird's weight, while UKn loggers weighed 15 g, or 1.2% of the bird's weight. E-obs loggers recorded GPS positions every hour in full battery mode and every 5 hr in normal battery mode. UKn loggers recorded GPS positions every hour with a full battery and every 12 hr with a normal battery. Both models operated continuously 24 hr a day. More detailed information about this dataset can be found in *Curk et al., 2022*.

The fieldwork was carried out in the Russian Arctic in 2013–2019 at four study sites: Kolguev Island (69°16′N, 48°87′E), Nenetsky Nature Reserve (68°20′N, 53°18′E), Vaigach Island (69°43′N, 60°08′E), and Yamal Peninsula (68°12′N, 68°59′E). For details on capture methods, see *Curk et al., 2022*; for detailed descriptions of study sites, see *Pokrovsky et al., 2015* for Kolguev Island and *Pokrovsky et al., 2019* for Yamal and Nenetsky.

During data pre-processing, we removed duplicated timestamps and calculated the mean daily positions of each individual. We partitioned the resulting dataset into several periods: (1) breeding, (2) fall migration, (3) first part of winter, (4) second part of winter, and (5) spring migration. We estimated the migration dates – the start and stop dates of the spring and fall migrations – using an iterative search procedure for piecewise regression described by *Crawley, 2007*. We created two models for each of the four dates: a simple model and a piecewise one. The simple model was linear, with latitude as the response variable and day of the year as a fixed effect. The piecewise model was also linear, with latitude as the response variable and day of the year as a fixed effect, and included two logical statements in its formula because we expected two linear segments in the fit. Specifically, it used 'Day Of the Year < BREAK' to define the left regression (before the estimated date) and 'Day Of the Year ≥ BREAK' for the right regression (after the estimated date). We explored different BREAK values to fit the piecewise model, looking for the value that minimized the residual standard error. The BREAK values examined included those associated with the start and end of the fall migration (September 5 to October 5 and October 5 to November 5, respectively), as well as those associated with the start and end of the spring migration (April 10 to May 10 and May 10 to June 10, respectively). We then validated (using the R function 'anova') that the piecewise model significantly improved the fit compared to the simple model (p<0.001). We estimated the date between winter's first and second parts as the day when the mean daily latitude was the lowest.

## Data analysis

First, we used linear mixed-effects models (R function 'lmer' in the library 'lme4'; *Bates et al., 2015*) to investigate whether or not Rough-legged buzzards migrated during winter. Latitude was the response variable, day of the year was a fixed effect, and individuals and year were included as random effects. Analyses were conducted separately for each migration period (fall, first phase of winter, second phase of winter, and spring). For both phases of the winter migration, we analyzed two additional models with longitude as the response variable instead of latitude. Likelihood ratio tests were used to compare candidate models. The year was not a calendar year but a year between two consecutive breeding seasons. Thus, fall migration, consecutive winter, and consecutive spring have the same value for the year. The day of the year was recalculated consecutively.

Second, we used linear mixed-effects models (R function 'lmer' in the library 'lme4') to investigate whether migrations' parameters differ between the migration periods. We analyzed four migration parameters: distance, duration, speed, and direction. The distance was calculated as the distance between two coordinates (start and end of migration) using the R function 'distm' in the library 'geosphere' (*Hijmans, 2016*). The duration was calculated as the number of days between the start and end of migration. Speed was calculated as the ratio of distance to duration. The direction was calculated as the bearing from the start of the migration coordinates to the end of the migration coordinates using the R function 'bearing' in the library 'geosphere' (*Hijmans, 2016*). Specific migration parameters (see below) were the response variables, the type of migration was a fixed factor, and individuals were a random factor. Likelihood ratio tests were used to compare candidate models. We considered four different parameters of migration (distance, duration, speed, and direction) and four types of migration (fall, first phase of winter, second phase of winter, and spring). The analysis was done separately for each of the migration parameters. Then, we used post hoc comparisons using the R function 'emmeans' in the library 'emmeans' (*Lenth et al., 2019*) to compare the estimated means.

In some raptor species, adult females disperse further than males (*Mearns and Newton, 1984*; *Serrano et al., 2001*; *Bildstein, 2006*; *Whitfield et al., 2009*). Therefore, we conducted an additional analysis on the effect of sex on migration length using linear mixed-effects models (R function 'lmer' in the library 'lme4'). The migration distance was used as the response variable, sex as a fixed factor, and individuals as a random factor.

Third, we investigated whether vegetation land cover differed between areas crossed during the quick (fall and spring) and slow (winter) migrations. We used the combined Terra and Aqua Moderate Resolution Imaging Spectroradiometer (MODIS) Land Cover Climate Modelling Grid (CMG) (MCD12C1) version 6 dataset (*Friedl and Sulla-Menashe, 2015*). We used a modified Leaf Area Index (LAI) as a classification scheme. We combined all four forest types and savannas into one category (forest) and excluded the categories: water bodies, and unclassified. We, therefore, had five types of vegetation cover: forest, grassland, cropland, shrubland, and urban. We annotated the mean daily positions with the vegetation cover type using the Env-DATA tool (*Dodge et al., 2013*). We used general linear mixed-effects models with a binomial distribution (R function 'glmer' in the library 'lme4') to investigate whether vegetation cover types differ between migration periods. Presence/absence of the studied vegetation land cover type was used as a response variable, migration type as a fixed factor, and individuals as a random factor. The analysis was done separately for each of the vegetation land cover types.

Fourth, we investigated whether snow cover could drive the slow migration phenomenon. We then compared the snow cover conditions the birds experienced during the winter with two hypothetical snow cover conditions that the birds would have experienced if they had not migrated during the winter. The first hypothetical snow cover condition would have happened if the birds had stayed where they arrived from the north (i.e., where their fall migration ended). To estimate this parameter, we calculated the winter dynamics of the average snow cover at the MCP occupied by the birds in October and April (northeast of their winter range). A second hypothetical snow cover condition would be if the birds flew immediately to the southwest and spent the whole winter there. To evaluate this, we calculated the winter dynamics of average snow cover on the MCPs occupied by the birds in January and February (southwest of their winter range). We then compared the values obtained for the real snow cover and two hypothetical snow covers using general linear mixed-effects models with a binomial distribution (R function 'glmer' in the 'lme4' library). Presence/absence of snow cover was used as a response variable, type of snow cover (real, first hypothetical, or second hypothetical) as a fixed factor, and years as a random factor. The analysis was done separately for each month. We then used post hoc comparisons to compare the estimated means, using the R function 'emmeans' in the 'emmeans' library (*Lenth et al., 2019*). We also tested whether the distance between two consecutive monthly MCPs during the over-wintering period is influenced by snow cover extent or snow cover difference. To investigate this, we employed a linear model, with the distance between two consecutive monthly MCPs as the response variable. Snow cover at the first MCP and the difference in snow cover between two consecutive MCPs were included as fixed factors.

We obtained monthly snow cover data with a spatial resolution of ca. 500 m (Global SnowPack MODIS) from the German Aerospace Center (DLR). This product is based on the Moderate Resolution Imaging Spectroradiometer (MODIS) daily snow cover products MOD10A1 and MYD10A1 (version 6 as provided by the National Snow and Ice Data Center NSIDC), which have been processed to remove the gaps due to cloud cover and polar darkness (*Dietz et al., 2015*). These processing steps include a combination of data available from different satellites (Aqua and Terra), 3-day temporal moving window filtering, a regional snow line elevation interpolation relying on a Digital Elevation Model, and a seasonal filter running through the time series for the whole hydrological year (1st of September through 31st August). The proportion of days in which one pixel is snow-covered per month is referred to here as fractional snow cover and is derived from these daily gap-filled rasters. Five MODIS tiles (h19v03, h20v03, h20v04, h21v03, and h21v04) were mosaicked and re-projected to WGS84. Then, for each month from October to April, we calculated 95% MCPs for the distribution of Rough-legged buzzards using the R function 'mcp' in library 'adehabitatHR' (*Calenge, 2006*). We extracted mean snow cover values from each MCP from every monthly snow cover raster separately using the R library 'raster' (*Hijmans, 2023*).

All calculations were performed using R version 4.2.2 'Innocent and Trusting' (*R Development Core Team, 2022*) and RStudio version 353 'Elsbeth Geranium' (*Posit team, 2022*).

## Discussion

Our study identified and characterized a bird migration pattern consisting of an alternation of quick (fall and spring migrations) and slow (over-wintering movements) phases. This migration pattern causes the over-wintering range of birds to shift and become dynamic throughout the season. This has obvious implications for range delineation and assessment of bird population trends.

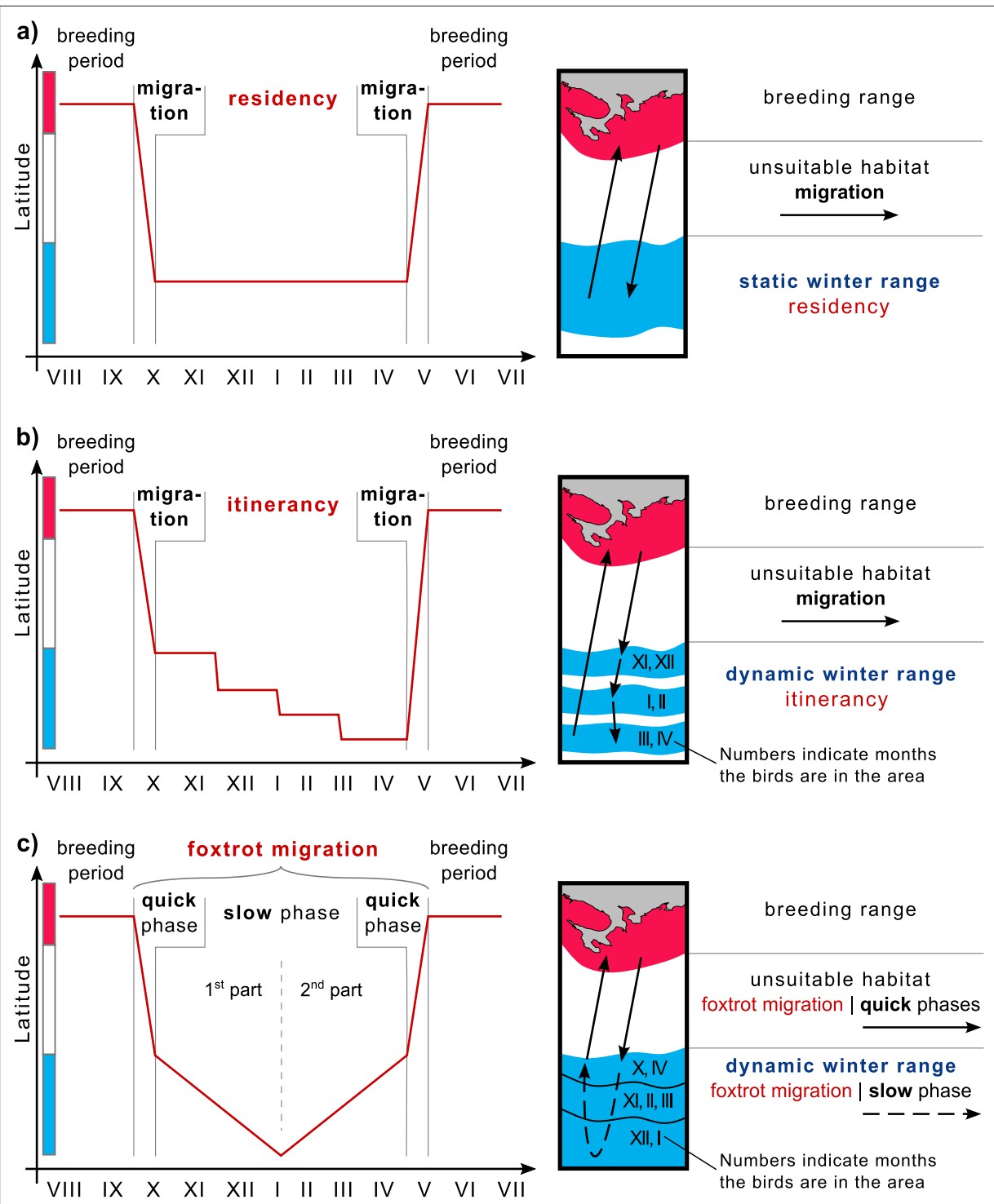

**Figure 4.** Winter strategies and over-wintering range scheme. (**a**) Residency, (**b**) Itinerancy, and (**c**) Foxtrot migration. The color bar along the y-axis corresponds to the color coding of the habitats of the chart on the right.

## Over-wintering movements and migratory patterns

Different bird species employ various wintering strategies, leading to diverse migration patterns. One strategy is 'residency', where birds arriving at over-wintering grounds remain there throughout the season, undertaking only minor foraging flights (*Figure 4a*). Their home ranges in over-wintering ranges are compatible with those in breeding ranges (*Kjellen et al., 1997*; *Alerstam et al., 2006*; *Newton, 2008*). Another strategy, initially identified by *Moreau, 1972*, is 'itinerancy', wherein birds fly between several sites separated by distances ranging from ten to several hundred kilometers throughout the over-wintering period, spending 1–2 months at each site (*Figure 4b*). Most Palearctic-African species adopt this itinerant behavior (*Trierweiler et al., 2013*; *Thorup et al., 2017*; *Schlaich et al., 2023*). The third strategy investigated here involves birds moving slowly and steadily throughout the over-wintering season, initially in one direction and then in the opposite direction. In 2009, Roine Strandberg and his colleagues used GPS transmitters to track Common buzzards (*Buteo buteo*). Despite some transmitters in their study ceasing to function mid-winter, the authors observed a phenomenon they termed 'prolonged autumn migration', wherein birds arriving at their wintering grounds did not halt migration but continued moving at a slower pace (*Strandberg et al., 2009*). In our study, we observed this phenomenon throughout the annual cycle. We demonstrated that, unlike itinerancy, movements in the over-wintering range are essentially a continuation of migration at a slower pace, influenced by external factors such as snow cover dynamics. Considering this behavior as an extension of migration and examining the complete life cycle of such species, it involves a quick phase during the transition between breeding and over-wintering ranges, followed by a slow phase of directed and seasonal movement within the over-wintering range, and finally, another quick phase toward the breeding range. Drawing an analogy to the alternating quick and slow movements of the foxtrot dance, we propose the term 'foxtrot migration' for this seasonal movement to offer a concise and easily understandable description (*Figure 4c*). Consequently, we suggest referring to the over-wintering range of species exhibiting itinerancy or foxtrot migration as the 'dynamic range' (*Figure 4b and c*).

Debates may arise regarding introducing a new term for this phenomenon and whether it should fall under the general term 'nomadism' or be categorized as a modified form of another migratory pattern, such as 'slow directional itinerancy'. We assert that behavioral pattern names should aid in understanding movement strategies and avoid confusion. One might ask whether it is necessary to distinguish between migration and wintering at all, given that migration is essentially a matter of tracking spatial and temporal changes in favorable conditions. Authors studying Montagu's harriers (*Circus pygargus*) argue that such generalization hampers understanding of wintering strategies (*Schlaich et al., 2023*). Instead, *Schlaich et al., 2023* propose explicitly considering the functions of animal-used sites, particularly their contributions to 'wintering' and 'refueling for migration'. For instance, cuckoos (*Canorus* sp.) utilize stopovers for both peak foraging conditions and refueling for long flights to subsequent sites (*Thorup et al., 2017*). Conversely, Montagu's harriers exhibit negligible refueling between sites due to short distances and energy-efficient soaring flight (*Trierweiler et al., 2013*; *Schlaich et al., 2023*). In our case, Rough-legged buzzards do not make several long stopovers, unlike species experiencing itinerancy, but move gradually in one direction and then return during the whole season (*Figure 1a*). As *Schlaich et al., 2023* suggested, examining the functions of animal-used sites reveals another difference between these strategies. The difference is that itinerant birds wait for better conditions at subsequent sites. In contrast, in our case, the southwestern areas that birds reach after several months are suitable and offer better conditions than the northeastern areas as they have less snow throughout the whole over-wintering season (*Figure 3*). All these differences highlight the discrepancy between the wintering strategies referred to as itinerancy and those described in this study. Therefore, using another term – foxtrot migration – for this phenomenon is more appropriate.

## The dynamic of the foxtrot migration

Our study affirmed the presence of foxtrot migration and a dynamic over-wintering range in Rough-legged buzzards. The taiga zone proved to be an unfavorable habitat during the quick phase of foxtrot migration (*Figure 2*), given the difficulty of Rough-legged buzzards locating open areas for hunting in this habitat. Conversely, the grassland and cropland zones served as favorable habitats during the slow phase throughout the entire over-wintering period (*Figure 2*), offering numerous open areas for

hunting. Within the grassland and cropland zones, the birds moved southwest and back over 1000 km throughout the winter (*Figure 1*). We did not observe a tendency for birds to leave over-wintering sites when snow cover reached a certain threshold (*Figure 3—figure supplement 1a*). Similarly, we found no evidence that birds stayed in areas with a certain amount of snow cover (*Figure 3*), nor did they leave sites when snow cover increased by a certain amount (*Figure 3—figure supplement 1b*).

We speculate that the studied birds, being experienced, anticipated that northeastern areas would become inaccessible later in the winter, prompting them to leave early without waiting for significant snow accumulation. Additionally, other factors, such as brief heavy snowfalls, might have triggered movement, even if these did not result in sustained increases in snow cover (*Vansteelant et al., 2011*). Alternatively, this phenomenon could have been more complex, with multiple factors possibly acting asynchronously, influencing their over-wintering movements (*Yanco et al., 2024*). While the ultimate reasons behind the over-wintering movements require further investigation, it is clear that snow cover was a proximate factor driving the birds' continual 1000 km southwest movement during winter (*Figure 3*). Our analysis revealed that if Rough-legged buzzards remained at their fall migration endpoint without moving southwest, they would encounter 14.4% more snow cover (99.5% vs. 85.1%, *Figure 3*). Although this difference may seem small (14.4%), it holds significance for rodent-hunting birds, distinguishing between complete and patchy snow cover. Complete snow cover makes hunting impossible, rendering these regions unsuitable for Rough-legged buzzards.

Unlike African-Palaearctic migrants exhibiting itinerant migratory behavior – waiting for suitable conditions in more southerly regions – the southern parts of the over-wintering ranges for Rough-legged buzzards remain favorable throughout the winter. If Rough-legged buzzards were to immediately fly to the southwest and stay there for the entire winter, they would experience 25.7% less snow cover (57.3% vs. 31.6%, *Figure 3b*). Despite this, it does not compel them to adopt this strategy. Thus, the observed phenomenon appears to be a trade-off between the desire to remain as close as possible to their breeding ranges and the search for suitable wintering conditions. Notably, one bird did follow this strategy by flying directly to the southwest and remaining there throughout the winter (*Figure 1—figure supplement 1*), exhibiting a residency strategy. This observation suggests that the distinction between winter strategies is flexible.

African-Palearctic migrants do not strictly adhere to a binary division between residency and itinerancy but demonstrate a continuum between these two strategies. Many species in Africa exhibit a mixture of both strategies in varying proportions; see *Schlaich et al., 2023* and reference therein. Similarly, in the mid-latitudes, species may not only show any of the three strategies but also a considerable degree of individual and regional variation. For example, Rough-legged buzzards from western Palearctic regions that overwinter in mid-latitude areas where snow cover dynamics are less pronounced than in eastern regions are likely to have a higher proportion of birds with a resident strategy. At the same time, Rough-legged buzzards in North America also exhibit a foxtrot migration pattern, although with variations. There, birds make a direct and rapid migration, usually across the boreal forest, followed by slower, more facultative movements once birds get south of the boreal. However, there is a lot of variation among individuals and even regions; there are more over-wintering slow movements in eastern North America but less in western North America (Neil Paprocki pers. comm.). Thus, Rough-legged buzzards in North America lie along a gradient between foxtrot migration and residency, likely influenced by different snow cover dynamics on the east and west coasts.

The foxtrot migration pattern is expected to be observed in many migratory species that exhibit distinct seasonal cycles in their over-wintering range. This migration pattern is expected to be prevalent for species living in regions where snow cover is a significant determinant of food availability, which is the case in large parts of the mid-latitudes. Therefore, understanding bird migration patterns is critical to accurately mapping ranges and assessing population trends in this region.

## Mapping ranges and assessing population trends

The implementation of our study is twofold: (1) the use of mid-winter bird surveys to determine over-wintering range may yield inaccurate results for species with dynamic range (*Figure 5a*), and (2) declines in abundance within a particular segment of the over-wintering range may indicate changes in range dynamics rather than widespread declines in species abundance (*Figure 5b*).

In North America and Europe, the number of over-wintering birds is typically estimated once a season in mid-winter. In North America, these estimates are made during the Christmas counts, usually

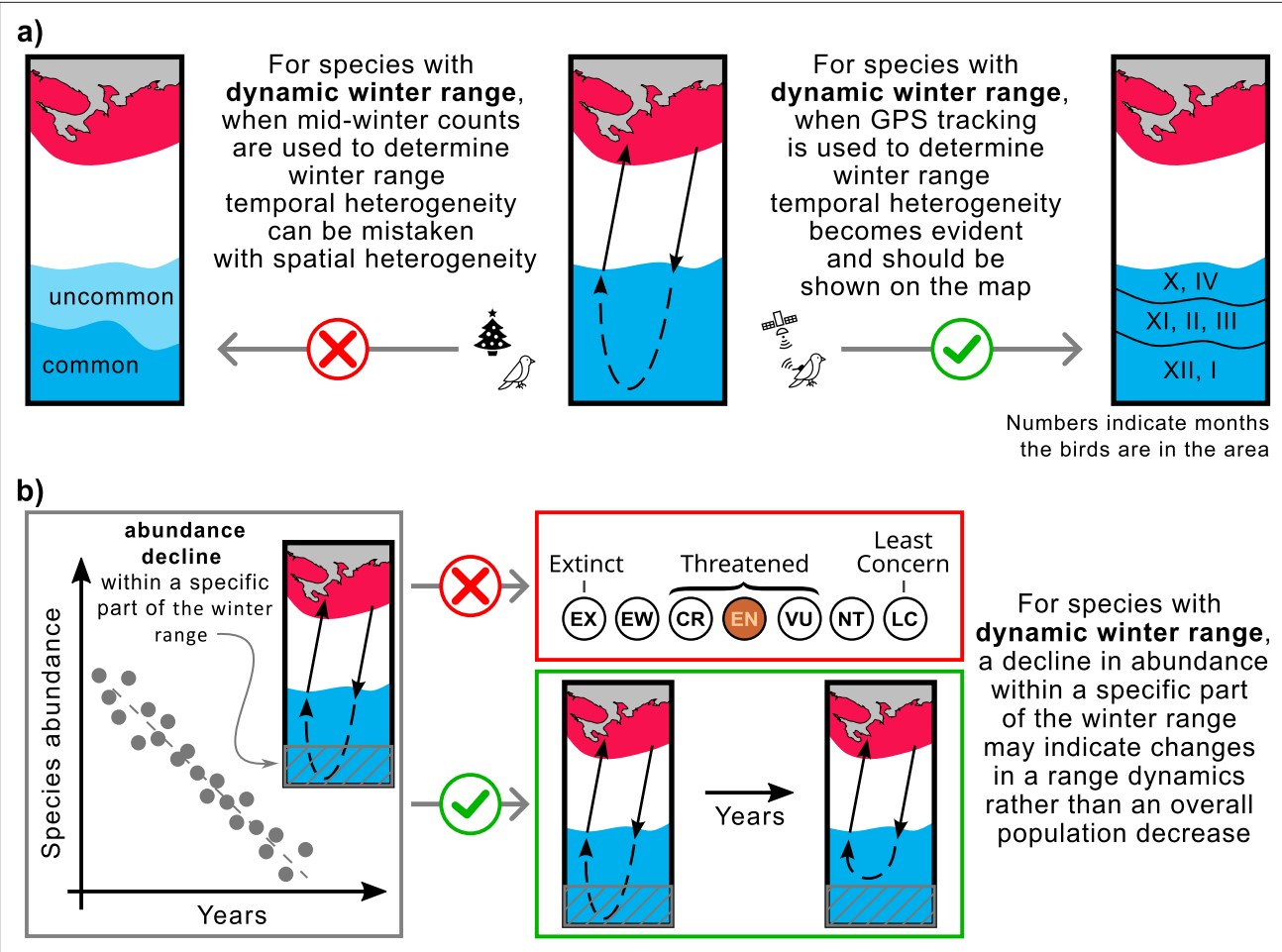

**Figure 5.** Mapping ranges and assessing population trends for the species with the dynamic over-wintering ranges. (**a**) Mid-winter counts, often used to determine winter range, may give a misleading representation of the over-wintering range for species with dynamic range. (**b**) Changing the conservation status of a species based on reduced abundance in a particular area may not be appropriate for species with dynamic range. Red, breeding range; blue, winter range.

once in late December (*National Audubon Society, 2024*), while in Europe, they are made during the IWC counts, usually once in January or February (*Wetlands International, 2024*). This approach can lead to errors in range estimation for birds with dynamic ranges. For such species, habitats occupied by birds in fall and spring will be listed as 'uncommon' at best, while habitats occupied by birds only in mid-winter will be listed as 'common'. However, the situation is the opposite if we consider the time birds spend in these habitats (*Figure 5a*). For example, based on our study of the Rough-legged buzzard, during mid-winter, the species is predominantly present in the southwestern portion of its over-wintering range, with only a tiny proportion present in the rest of the region. As a result, a map of the over-wintering range may show the species as 'common' in the southwest and 'uncommon' in the northeast. This map would be inaccurate because, during the entire over-wintering period, Rough-legged buzzards spend both fall and spring in the northeastern part and only mid-winter in the southwestern part.

To address this, continuous year-round GPS tracking of the species provides a means to track bird locations throughout the over-wintering season, facilitating the creation of accurate distribution maps, particularly for species that exhibit dynamic ranges. We advocate representing temporal heterogeneity (range dynamics) on maps as distinct zones, denoting periods when the species is abundant in a given area. To distinguish temporal from spatial heterogeneity, we recommend using lines to delineate the boundaries of these zones, rather than color shading, and incorporating numbers to denote

the months of species abundance in a given zone (*Figure 5a*). We suggest ecologists include dynamic over-wintering ranges in descriptions and range maps for foxtrot migratory and itinerant bird species.

Population counts for a species are often limited to a portion of its range. Therefore, conclusions about conservation status drawn from such counts may be misleading. A decline in abundance within a particular portion of the over-wintering range may indicate changes in range dynamics rather than a general decline in the species (*Figure 5b*). For example, climate change may affect snow cover dynamics, reducing its intensity in northern regions. As a result, species whose range dynamics depend on snow cover may choose to remain in the northern areas and not migrate as far south as they traditionally have. Despite this shift in range dynamics, overall species abundance may remain unchanged. Similar patterns have affected Rough-legged buzzards in some areas of the European over-wintering range.

A 2022 Dutch study found a decline in wintering Rough-legged buzzards over the last 40 years (*Hornman and Boele, 2022*). On the one hand, this may represent a conservation concern. On the other hand, applying the rationale of the dynamic range, the apparent local decline may be attributed to climate change, resulting in less comprehensive snow coverage in the northeastern wintering areas of Rough-legged buzzards relative to the Netherlands. Such a shift in snow coverage makes it less probable for the birds to migrate to the Netherlands for over-wintering. This proposition is further supported by a study of the winter population dynamics of Rough-legged buzzards in the Netherlands in 2011, showing that the main winter population peak occurred in late December, with many birds migrating (*Vansteelant et al., 2011*). *Vansteelant et al., 2011* also found that the main migration occurred after heavy snowfall in northern Europe, supporting our foxtrot migration explanation for this decline. Therefore, investigating the dynamic range is critical to understanding a species' range and effectively assessing its conservation status.

## Conclusion

Our study sheds light on the over-wintering strategy of the Rough-legged buzzard, an Arctic breeder that migrates to mid-latitudes during the non-breeding season. We identified a distinct migration pattern where, following an initial 1500 km journey to mid-latitudes, the birds continue to move slowly southwestward and then back northeastward throughout the entire over-wintering season. This movement, covering 1000 km in one direction, makes their over-wintering range dynamic. We call this migration pattern 'foxtrot migration', drawing an analogy with the alternating quick and slow steps of the foxtrot dance. Our results highlight the importance of this migration pattern and the concept of 'dynamic range' in revealing potential inaccuracies in range mapping from single midwinter surveys and suggest that species distribution maps should represent temporal variations in range dynamics as separate zones. Understanding and accounting for this migratory behavior is critical for accurately assessing bird populations in over-wintering ranges. This is especially important for species such as Arctic nesting birds, where population assessments are primarily conducted in over-wintering ranges due to the logistical and political challenges of the Arctic.

## Acknowledgements

We are very grateful to everyone who helped us collect data in the field. We would like to thank Dr Theunis Piersma and two anonymous reviewers for their valuable feedback and constructive criticism, which significantly improved the quality of this publication. This study was funded by the Max-Planck Institute of Animal Behavior and the German Air and Space Administration (DLR). We also acknowledge partial funding by the Deutsche Forschungsgemeinschaft (DFG, German Research Foundation) under Germany's Excellence Strategy – EXC 2117-422037984.

## Additional information

### Funding

| Funder | Grant reference number | Author |
|---|---|---|
| Deutsche Forschungsgemeinschaft | EXC 2117 - 422037984 | Ivan Pokrovsky Martin Wikelski |
| Max Planck Institute of Animal Behavior | | Ivan Pokrovsky Martin Wikelski |
| Deutsches Zentrum für Luft- und Raumfahrt | | Ivan Pokrovsky Andreas Dietz Sebastian Rößler Martin Wikelski |

The funders had no role in study design, data collection and interpretation, or the decision to submit the work for publication.

### Author contributions

Ivan Pokrovsky, Conceptualization, Resources, Data curation, Formal analysis, Validation, Investigation, Visualization, Methodology, Writing - original draft, Project administration, Writing - review and editing; Teja Curk, Formal analysis, Investigation, Writing - review and editing; Andreas Dietz, Formal analysis, Funding acquisition, Investigation, Methodology, Writing - review and editing; Ivan Fufachev, Formal analysis, Investigation, Methodology, Writing - review and editing; Olga Kulikova, Investigation, Writing - review and editing; Sebastian Rößler, Resources, Data curation, Formal analysis, Investigation, Methodology, Writing - review and editing; Martin Wikelski, Resources, Data curation, Formal analysis, Supervision, Funding acquisition, Validation, Investigation, Methodology, Project administration, Writing - review and editing

### Author ORCIDs

Ivan Pokrovsky https://orcid.org/0000-0002-6533-674X

### Ethics

IP applied for and obtained permit No. 77-18/0854/4388 from The General Radio Frequency Centre, permit No. RU/2018/406 from Federal Service for Supervision of Communications, Information Technology and Mass Media (Roskomnadzor), and permit No. RU0000045099 from Federal Security Service. No specific permissions were required from Federal Service for Supervision of Natural Resources (Rosprirodnadzor) according to Section 44 and Section 6 of the Federal Law of the Russian Federation No. 52 from 24.04.1995 (last update 08.08.2024) On Wildlife, and from Federal Service for Technical and Export Control (FSTEC/FSTEK) according to Russian Federation government decree No. 633 from 29.08.2001 and Letter from FSTEK No. 240/33/1373 from 06.04.2015. There were no Special Protected Natural Territories in our study area, and our activities did not include the withdrawal of investigated species from nature. All our protocols met the ABS/ASAB guidelines for the ethical treatment of animals. In Nenetsky, the work was carried out in agreement with the Nenetsky Nature Reserve in a buffer zone.

Reviewer #4 (Public review): https://doi.org/10.7554/eLife.87668.4.sa1
Author response https://doi.org/10.7554/eLife.87668.4.sa2

## Additional files

### Supplementary files

• Supplementary file 1. Supplementary tables. (a) Table S1. The relationship between latitude/longitude and the day of the year during quick and slow phases of migration. The likelihood ratio test compares two candidate models: with ('~doy') and without the day of the year ('~1') as a fixed factor. (b) Table S2. The relationship between latitude/longitude and the day of the year during quick and slow phases of migration. Linear mixed-effect model, fixed effects. The response variable – latitude/longitude. Fixed effect – the day of the year ('doy'). Random effects – individuals and

year. (c) Table S3. The relationship between latitude/longitude and the day of the year during quick and slow phases of migration. Linear mixed-effect model, random effects. The response variable – latitude/longitude. Fixed effect – the day of the year. Random effects – individuals ('bird') and year ('year'). (d) Table S4. The difference between the distance of slow and quick migrations. Linear mixed-effect model, post hoc results. The response variable – distance (km). Fixed effect – the type of migrations. Results of the post hoc comparison. (e) Table S5. The difference between the duration of slow and quick migrations. Linear mixed-effect model, post hoc results. The response variable – duration (days). Fixed effect – the type of migrations. Results of the post hoc comparison. (f) Table S6. The difference between the speed of slow and quick migrations. Linear mixed-effect model, post hoc results. The response variable – speed (km/day). Fixed effect – the type of migrations. Results of the post hoc comparison. (g) Table S7. The difference between the direction of the spring and the second phase of the winter migration and between the autumn and the first phase of the winter migration (two models). Linear mixed-effect models, post hoc results. The response variable in both models – direction (deg). Fixed effect – the type of migrations. Results of the post hoc comparison. (h) Table S8. The relationship between migration distance and the sex of the birds. The likelihood ratio test compares two candidate models: with ('~sex') and without the sex ('~1') as a fixed factor. (i) Table S9. The difference between vegetation land cover types crossed during quick (fall and spring) and slow (winter) migrations. General linear mixed-effect models. Results are given on the logit scale. (j) Table S10. The difference between snow cover conditions in the real situation ('Real') and two and two hypothetical situations – if birds spend winter in the place where they have arrived after fall migration ('Hyp stay') and if birds fly directly to the Southwest and stay there all winter ('Hyp SW'). General linear mixed-effect models. Results are given on the logit scale. Results of the post hoc comparison.

- MDAR checklist

### Data availability

Tracking data have been deposited in the Movebank Data Repository.

The following previously published dataset was used:

| Author(s) | Year | Dataset title | Dataset URL | Database and Identifier |
|---|---|---|---|---|
| Pokrovsky I, Kulikova O, Wikelski M | 2021 | Data from: Longer days enable higher diurnal activity for migratory birds [rough-legged buzzards] | https://datarepository.movebank.org/handle/10255/move.1289 | Movebank Data Repository, 10.5441/001/1.dg3sm625 |

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
