## [Editor Report · eLife assessment]

This **fundamental** work describes an understudied bird migration pattern using data from an Arctic raptor. With an extensive dataset and comprehensive analyses, the observed pattern is **convincing**. This study will be of interest to researchers exploring the ecological drivers of bird migration.

---

## [Referee Report · Reviewer #4 (Public review)]

Summary:

This study describes an understudied migration pattern of dynamic non-breeding range using data from an Arctic raptor. Using data from GPS tags, the study describes the known pattern of fast migration during autumn and spring, and an undescribed pattern of slow migration, at much slower pace, throughout the over-wintering season.

Strengths:

The study presents a comprehensive analysis of the annual cycle of an interesting and undescribed migration system. The conceptual advancement is original and the data is rich and persuading. The Discussion part of the manuscript is well written.

Weaknesses:

Other sections of the manuscript need some more polish, both in terms of the terminology, the language and the logic of the presentation of the subject. The title is not good. During most of the text, the authors do not properly follow a certain terminology regarding migration, over-wintering, non-breeding range, and this is very confusing. So, consistency of the text is warranted. A bigger issue is the selection of latitudes (or the actual reason for movement) during the over-wintering period. The study claims that this relates to snow cover but fails to properly demonstrate it. It is likely that the birds move because of changes in snow cover rather than because of the level of snow cover. This is a testable prediction. A possible explanation is that there is a cost for moving further south and thus the birds are reluctant of moving unless they are forced to do it by the high snow cover. Another, similar and testable prediction is that the birds aim at selecting latitudes where snow cover is partial and move slowly during the winter to areas that are only partially covered by the snow with the progression of the winter. A modified, non-linear, snow cover analysis using GAMM could uncover such patterns.

---

## [Author Response]

The following is the authors’ response to the current reviews.

**Reviewer #4:**

We sincerely appreciate the time and effort you have taken to review our manuscript. We followed your recommendations to polish the text and make it easier to understand.

Regarding terms and terminology, we changed “non-breeding” everywhere in the text to “over- wintering.”

Regarding the title, as it was suggested by reviewer #1 as his recommendation, we tried to find a compromise and make the changes you suggested but left part of the suggestion from reviewer #1. So, now it’s “Foxtrot migration and dynamic over-wintering range of an arctic raptor”

Thank you for highlighting the importance of snow cover and changes in snow cover as a possible factor of over-wintering movements. We appreciate your feedback and have explored several approaches to address this issue. Specifically, we examined how both snow cover extent and changes in snow cover influenced movement distance. However, we found no effect of either factor on movement distance.

Our data show that birds leave their sites in October and move southwest, even though snow cover is minimal at that time. They also leave their sites in November and in subsequent months, regardless of the snow cover levels. Thus, we observed no pattern of birds leaving sites when snow cover reaches a specific threshold (e.g., 75-80%). Similarly, we found no evidence of birds staying in areas with a certain snow cover extent (e.g., 30%), nor did they leave sites when snow cover increased by a specific amount (e.g., by 10 or 20%).

It is possible that more experienced birds anticipate that October plots will become inaccessible later in the winter and, therefore, leave early without waiting for significant snow accumulation. Alternatively, other factors, such as brief heavy snowfalls, may trigger movement, even if these do not lead to sustained increases in snow cover. Multiple factors, possibly acting asynchronously, could also play a role. This complexity adds an interesting dimension to the study of ecological patterns. However, in this study, we chose to focus on describing the migration pattern itself and its impact on aspects like over-winter range determination and population dynamics. While we have prioritized this approach, we remain committed to further analyzing the data to uncover additional details about this behavior.

In response to your suggestion, we have expanded the Methods sections to clarify that we tested the effects of snow cover and changes in snow cover on distance (Lines 241-246); the Results section (Lines 348-349). We have also included the relevant plots in the Supplementary Materials. In the Discussion, we noted that this approach did not reveal any significant dependence and acknowledged that this issue requires further investigation (Lines 422-459).

The following is the authors’ response to the previous reviews.

**Reviewer #2:**

We sincerely appreciate the time and effort you have taken to review our manuscript.

First of all, we apologize for publishing the preprint without incorporating certain adjustments outlined in our earlier response, particularly in the Methods section. This was due to an oversight regarding the different versions of the manuscript. We have corrected this mistake. Our response to the feedback on this section (Methods), with line numbers of the changes made, is immediately below this response. In addition, we have included the units of measurement (mean and standard deviation) in both the results and figure captions for clarity.

To focus on the main point regarding wintering strategies, we acknowledge that in the previous versions, this aspect was inadequately addressed and caused some confusion. In the revised edition, both the Introduction and the Discussion have been thoroughly reworked.

As you suggested, we have removed the long introductory paragraph and all references to foxtrot migrations from the Introduction. As a result, the Introduction is now short and to the point. In the second paragraph, we explain why we propose the wintering strategies outlined (L74-81).

In the Discussion, we've added a substantial new section at the beginning that discusses different wintering strategies. We have also updated Figure 4 accordingly. Previously, we erroneously suggested that Montagu's harrier and other African-Palaearctic migrants might adopt wintering strategies similar to those we describe. Upon further investigation, however, we found that almost all African-Palaearctic migrants exhibit an itinerant wintering strategy. Conversely, the strategy we describe is primarily observed in mid-latitude wintering species.

We have shown that, unlike itinerancy, the birds in our study don't pause for 1-2 months at multiple non-breeding sites, but instead migrate significant distances, up to 1000 km, throughout the winter. Furthermore, unlike itinerancy, the sites they reach are consistently snow-free throughout the year. Following the logic of publications on Montagu's harriers (Schlaich et al. 2023), our birds do not wait for favorable conditions at the next site, as is typical of itinerancy. Moreover, this behavior is influenced by external factors such as snow cover dynamics and occurs primarily in mid-latitudes. Researchers studying a species similar to our subject, the Common buzzard, observed a similar pattern and termed it "prolonged autumn migration" rather than itinerancy. Although their transmitters stopped working in mid-winter, precluding a full observation of the annual cycle, they captured the essence of continued migration at a slower pace, distinct from itinerancy. We've detailed all of these findings in a new section.

In addition, we acknowledge the mischaracterization of the implications of our research as ‘Conservation implications’ and have corrected this to ‘Mapping ranges and assessing population trends’, as you suggested.

Finally, we've rewritten the Conclusion, removing overly grandiose statements and simply summarizing the main findings.

We appreciate your time and effort in reviewing our manuscript. With your invaluable input, it has become clearer, more concise, and easier to understand.

Dataset: unclear what is the frequency of GPS transmissions. Furthermore, information on relative tag mass for the tracked individuals should be reported.

We have included this information in our manuscript (L 115-122). We also refer to the study in which this dataset was first used and described in detail (L 123).

Data pre-processing: more details are needed here. What data have been removed if the bird died? The entire track of the individual? Only the data classified in the last section of the track? The section also reports on an 'iterative procedure' for annotating tracks, which is only vaguely described. A piecewise regression is mentioned, but no details are provided, not even on what is the dependent variable (I assume it should be latitude?).

Regarding the deaths, we only removed the data when the bird was already dead. We estimated the date of death and excluded tracking data corresponding to the period after the bird's death. We have corrected the text to make this clear (L 130-131).

Regarding the piecewise regression. We have added a detailed description on lines 136-148.

Data analysis: several potential issues here:(1) Unclear why sex was not included in all mixed models. I think it should be included.

Our dataset contains 35 females and eight males (L116). This ratio does not allow us to include sex in all models and adequately assess the influence of this factor. At the same time, because adult females disperse farther than males in some raptor species, we conducted a separate analysis of the dependence of migration distance on sex (Table S8) and found no evidence for this in our species. We have written about that in the Methods (L177-181) and after in the Results (L277-278).

(2) Unclear what is the rationale of describing habitat use during migration; is it only to show that it is a largely unsuitable habitat for the species? But is a formal analysis required then? Wouldn't be enough to simply describe this?

Habitat use and snow cover determine the two main phases (quick and slow) of the pattern we describe. We believe that habitat analysis is appropriate in this case, and a simple description would be uninformative and not support our conclusions.

(3) Analysis of snow cover: such a 'what if' analysis is fine but it seems to be a rather indirect assessment of the effect of snow cover on movement patterns. Can a more direct test be envisaged relating e.g. daily movement patterns to concomitant snow cover? This should be rather straightforward. The effectiveness of this method rests on among-year differences in snow cover and timing of snowfall. A further possibility would be to demonstrate habitat selection within the entire non-breeding home range of an individual in relation snow cover. Such an analysis would imply associating presenceabsence of snow to every location within the non-breeding range and testing whether the proportion of locations with snow is lower than the proportion of snow of random locations within the entire nonbreeding home range (95% KDE) for every individual (e.g. by setting a 1/10 ratio presence to random locations).

The proposed analysis will provide an opportunity to assess whether the Rough-legged buzzard selects areas with the lowest snow cover, but will not provide an opportunity to follow the dynamics and will therefore give a misleading overall picture. This is especially true in the spring months. In March-April, Rough-legged buzzards move northeast and are in an area that is not the most open to snow. At this time, areas to the southwest are more open to snow (this can be seen in Figure 3b). If we perform the proposed analysis, the control points for this period would be both to the north (where there is more snow) and to the south (where there is less snow) from the real locations, and the result would be that there is no difference in snow cover.

A step-selection analysis could be used, as we did in our previous work (Curk et al 2020 Sci Rep) with the same Rough-legged buzzards (but during migration, not winter). But this would only give us a qualitative idea, not a quantitative one - that Rough-legged Buzzards move from snow (in the fall) and follow snowmelt progression (in the spring).

At the same time, our analysis gives a complete picture of snow cover dynamics in different parts of the non-breeding range. This allows us to see that if Rough-legged buzzards remained at their fall migration endpoint without moving southwest, they would encounter 14.4% more snow cover (99.5% vs. 85.1%). Although this difference may seem small (14.4%), it holds significance for rodent-hunting birds, distinguishing between complete and patchy snow cover.

Simultaneously, if Rough-legged buzzards immediately flew to the southwest and stayed there throughout winter, they would experience 25.7% less snow cover (57.3% vs. 31.6%). Despite a greater difference than in the first case, it doesn't compel them to adopt this strategy, as it represents the difference between various degrees of landscape openness from snow cover.